# A Randomized, Crossover Study of the Acute Cognitive and Cerebral Blood Flow Effects of Phenolic, Nitrate and Botanical Beverages in Young, Healthy Humans

**DOI:** 10.3390/nu12082254

**Published:** 2020-07-28

**Authors:** Philippa A. Jackson, Emma L. Wightman, Rachel Veasey, Joanne Forster, Julie Khan, Caroline Saunders, Siobhan Mitchell, Crystal F. Haskell-Ramsay, David O. Kennedy

**Affiliations:** 1Brain Performance and Nutrition Research Centre, Northumbria University, Newcastle Upon Tyne NE1 8ST, UK; philippa.jackson@northumbria.ac.uk (P.A.J.); emma.l.wightman@northumbria.ac.uk (E.L.W.); rachel.veasey@northumbria.ac.uk (R.V.); jo.forster@northumbria.ac.uk (J.F.); julie.khan@northumbria.ac.uk (J.K.); crystal.haskell-ramsay@northumbria.ac.uk (C.F.H.-R.); 2NUTRAN, Northumbria University, Newcastle Upon Tyne NE1 8ST, UK; 3PepsiCo, Nutrition Sciences Global R&D, 700 Anderson Hill Rd, Purchase, NY 10577, USA; caroline.saunders@suntory.com (C.S.); siobhan@noom.com (S.M.)

**Keywords:** cerebral blood flow, near-infrared spectroscopy, polyphenols, dietary nitrate

## Abstract

Background: In whole foods, polyphenols exist alongside a wide array of other potentially bioactive phytochemicals. Yet, investigations of the effects of combinations of polyphenols with other phytochemicals are limited. Objective: The current study investigated the effects of combining extracts of beetroot, ginseng and sage with phenolic-rich apple, blueberry and coffee berry extracts. Design: This randomized, double-blind, placebo-controlled crossover design investigated three active beverages in 32 healthy adults aged 18–49 years. Each investigational beverage comprised extracts of beetroot, ginseng and sage. Each also contained a phenolic-rich extract derived from apple (containing 234 mg flavanols), blueberry (300 mg anthocyanins) or coffee berry (440 mg chlorogenic acid). Cognition, mood and CBF parameters were assessed at baseline and then again at 60, 180 and 360 min post-drink. Results: Robust effects on mood and CBF were seen for the apple and coffee berry beverages, with increased subjective energetic arousal and hemodynamic responses being observed. Fewer effects were seen with the blueberry extract beverage. Conclusions: Either the combination of beetroot, ginseng and sage was enhanced by the synergistic addition of the apple and coffee berry extract (and to a lesser extent the blueberry extract) or the former two phenolic-rich extracts were capable of evincing the robust mood and CBF effects alone.

## 1. Introduction

Emerging data from human intervention studies suggest that polyphenol-rich foods and beverages may be able to deliver acute benefits to cardiovascular function [1,2] and cognitive performance [3,4,5,6,7,8]. The brain function effects of polyphenols (e.g., flavanols and anthocyanins) may well be related, in part, to their vasodilatory properties [9,10]. Here, polyphenols are able to increase endogenous production of the potent vasodilator nitric oxide (NO) [11,12]. However, demonstrations of concurrent modulation of cerebral blood flow (CBF) and either cognitive function or psychological state have been elusive [5,13,14]. 

In whole foods, polyphenols are found alongside a wide range of other plant-derived bioactive compounds and yet, to date, very little research exists on the co-administration of phytochemicals. Where data do exist, we see that combinations of phytochemicals can reveal synergistic effects [15,16,17]. With regard to co-administration of polyphenols with other phytochemicals, those with similar or complementary putative mechanisms may be of most interest. For example, in addition to their direct effects on NO production, polyphenols may also catalyze the reduction of nitrite to NO [18]; suggesting a potential additive effect on NO production when consuming nitrates and polyphenols in combination. In support of this, while Kennedy et al. [14] found that CBF changes were not accompanied by improved cognitive performance 45 min following administration of the stilbene polyphenol resveratrol (even though this represented the T_max_ of plasma metabolite levels; this being broadly in line with the T_max_ of flavanol polyphenols (e.g., epigallocatechin gallate (EGCG; 78–162 min post-dose) and phenolic acids (36–138 min post-dose [19])), a similar methodology by the same lab, using beetroot juice (a source of dietary nitrate and polyphenols [20]) did demonstrate both improved cognitive performance and modulation of hemodynamic responses during post-dose task performance. This commenced 90 min after treatment and was in conjunction with significantly increased levels of plasma nitrite [21].

Although less well explored, certain constituents of herbal extracts such as sage and ginseng also modulate NO synthesis. This may, in part, underpin some of their observed beneficial effects on cognition and mood in acute human intervention studies [22,23,24,25]. For example, ginsenosides Rg1 and Rb1 have been shown to induce phosphorylation of endothelial nitric oxide synthase (eNOS), resulting in increased NO production [26]. With regard to sage, the reported beneficial effects of this herbal extract on cognition are thought to be driven by the cholinesterase-inhibiting properties of its terpenoid constituents; of which, 1,8-cineole may be the most potent [27]. Interestingly, 1,8-cineole has also been shown to act as a vasorelaxant; which may be driven by its actions on NO production [28]. Neither sage nor ginseng can boast a great deal of pharmacokinetic information and only a very small proportion of this is derived from human participants following oral consumption. Nevertheless, two small trials demonstrate that 1,8-cineole (sage) and 20(R)-ginsenoside Rg3 (metabolite of ginseng) can achieve a T_max_ of ~45 and 40 min post-oral consumption of 6.4 g sage brewed into a tea and 3.2 mg/kg ginseng, respectively [29,30].

Given that polyphenols, beetroot, ginseng and sage share a putative common mechanism relevant to brain function—namely modulation of NO synthesis—the objective of this randomized, double-blind, crossover study was to explore the cognitive, mood and CBF effects of single doses of three investigational drinks. Each of these contain extracts of beetroot, sage and ginseng, but with the addition of separate phenolic-rich extracts derived from apple (rich in flavanols), blueberry (rich in anthocyanins) and coffee berry (rich in chlorogenic acids). We hypothesize that co-administration of these phenolics with beetroot, ginseng and sage will lead to differential effects on the behavioral and CBF parameters. Given the diverse range of polyphenol structural groups, the separate phenolic-rich fruit extracts were administered to investigate potential variability in efficacy of the different combinations based on the individual phenolic profile of the extracts. The time course of assessments here is somewhat exploratory, given the small amount of pharmacokinetic data available for all pertinent active compounds in humans following oral administration, and is fairly broad (60, 180 and 360 min post-dose) in order to mirror the T_max_ of the predominant actives noted above and to extend the possible window of opportunity.

## 2. Subjects and Methods

### 2.1. Study Design and Participants

The study followed a randomized, double-blind, placebo-controlled crossover design with four study arms. It was conducted from August 2014 to March 2015 at the Brain, Performance and Nutrition Research Centre (BPNRC), Northumbria University. Northumbria University’s Health and Life Sciences Ethics Committee reviewed and approved all study procedures (code: RE-HLS-13-140414-534bda035fe3b) and the investigations were carried out following the rules of the Declaration of Helsinki of 1975. The study was preregistered with Clinical trials.gov (reference: NCT02202629).

A convenience sample of 32 adults aged 18 to 35 years completed the study, recruited from within the staff and student population of Northumbria University (*n =* 6 males, 26 females, mean age 22.28 years (SD 4.27), with a mean BMI of 23.43 kg/m^2^, and a mean blood pressure of 119.42/77.03 mmHg, *n* = 28 right handed). Participants declared themselves to be in good health, have normal or corrected-to-normal vision and English as their first language. Exclusion criteria were a BMI of < 18 or > 35 kg/m^2^, high blood pressure (defined as systolic > 139 mmHg or diastolic > 89 mmHg), smoking, food allergies or insensitivities, pregnancy, breast feeding, currently taking any medication (use of contraceptives was not excluded) or dietary supplements, sleep disturbances and/or taking sleep aid medication, history of neurological, vascular or psychiatric illness, current diagnosis of anxiety or depression, migraines, recent history (within 12 months) of alcohol/substance abuse, disorder of the blood, heart disorder/history of vascular illness, respiratory disorder requiring regular medication, Type I or II diabetes, renal disease, hepatic disease, severe disease of the gastrointestinal tract and any health condition that would prevent the fulfilment of the study requirements. BMI and blood pressure were measured in the laboratory; all other exclusion criteria were self-reported.

Sample size estimation was made assuming a medium effect size seen in previous similar studies [4,14,31]. Using this estimate, the sample sizes for an entirely within-subjects design incorporating four intervention groups with 0.5 correlation between repeated measures required to achieve power of 0.8 at α = 0.05 varied between 28 participants (for the CDB cognitive measures) and 20 participants (for the NIRS measurements). The sample size was therefore set at 32 participants to allow for a full Latin square design. 

### 2.2. Treatments

On each of the four test visits, participants received a different beverage by random allocation according to a counterbalancing order (Latin square) generated by computer. All beverages were administered in opaque containers by an independent third party who had no other involvement in the study. The treatments were a placebo cherry-flavored beverage sweetened with sucralose (4%) and three investigational products. The placebo contained no active ingredients and the cherry flavor was selected to mask the flavor of the other ingredients. Each of the investigational products comprised the placebo drink, plus 10 g beetroot extract (1.5% nitrate), 170 mg ginseng extract (4.5% ginsenosides) and 280 mg sage extract plus one of three polyphenol containing extracts—either 2.49 g blueberry extract (300 mg blueberry anthocyanins), 275 mg apple extract (234 mg flavanols expressed as epicatechin equivalents) or 1.1 g coffee berry extract (440 mg chlorogenic acid). See Table 1.

### 2.3. Cognitive and Mood Assessment

All cognitive function tests were delivered using the Computerized Mental Performance Assessment System (COMPASS, Northumbria University, Newcastle upon Tyne, UK). This testing system delivers a tailor-made collection of tasks, with fully randomized parallel versions of each task delivered at each assessment for each individual. 

The cognitive assessment comprised four repetitions of the ten-minute battery of three tasks that make up the Cognitive Demand Battery (CDB) plus mental fatigue and alertness visual analogue scale ratings, resulting in a total of 40 min performance of cognitively demanding tasks per assessment. The CDB comprises serial subtraction (3 s/7 s) and focused attention (rapid visual information processing [RVIP]) tasks, described in full elsewhere [21]. This approach, inculcating high cognitive demands for an extended period of time, has been shown to be sensitive to the effects of numerous nutritional interventions [4,21,25,32].

Given that executive functioning has been shown to be sensitive to mental fatigue, two executive function tasks (Stroop task, Peg and Ball) were also performed immediately after the CDB. In addition, given the potential for flavonoids to modulate hippocampal function, an assessment of episodic memory was incorporated into the battery of tasks and included immediate and delayed word recall, word recognition and picture recognition. Similar selections of tasks delivered using COMPASS have been shown to be sensitive to a number of nutritional interventions [33,34,35,36]. Further, these tasks have been shown to activate the prefrontal cortex, the area of interest with regards to the NIRS measurements [14,37,38]. Full descriptions of the above COMPASS tasks can be found in Appendix A and the CDB in Appendix A.

Mood/psychological state was assessed with two well-validated measures of mood including the Bond–Lader visual analogue mood scales [39] and the Profile of Mood States (POMS) questionnaire [40]. The Bond–Lader mood scales comprise sixteen lines anchored at either end by antonyms (e.g., ‘alert-drowsy’ and ‘calm-excited’). Participants indicate their current subjective position between the antonyms on the line and ratings are scored as % along the line from left to right. Outcomes comprise three factor analysis-derived scores: ‘Alertness’, ‘Calmness’ and ‘Contentment’. For the POMS, scores on six dimensions are generated (tension/anxiety, anger/hostility, fatigue/inertia, depression/dejection, confusion/bewilderment, vigor/activity), and an overall score termed ‘Total Mood Disturbance (TMD)’ can then be calculated by summing the five negative subscale scores and subtracting the vigor score. A higher score on TMD indicates a greater degree of mood disturbance. 

In order to further assess participants’ subjective response to the treatments, four in-house visual analogue scales (presented as lines, as above) were also included: ‘How focused do you feel right now?’, ‘How clear is your mind right now?’, ‘How productive do you feel right now?’, ‘How do you rate your ability to solve a problem right now?’ The latter scale was anchored by the endpoints ‘very bad’ (left-hand end) and ‘very good’ (right-hand end). The other scales had the endpoints ‘not at all’ (left-hand end) and ‘extremely’ (right-hand end).

The running order of the cognitive tasks is presented in Figure 1. The entire assessment took 60 min to complete.

### 2.4. NIRS

Hemodynamic response was monitored using a frequency domain ‘quantitative’ NIRS system (OxiplexTS Frequency-Domain Near-Infrared Tissue Oximeter, ISS Science). This system gives absolute measurements of the absorption of near-infrared light emitted at two distinct wavelengths, which allows for the quantification of oxygenated hemoglobin (HbO_2_) and deoxygenated hemoglobin (HHb) [41]. These values are then used to determine total hemoglobin (tHb = HbO_2_ + HHb) and oxygen saturation (StO_2_ = HbO_2_/tHb × 100%). This system is ideal for quantifying acute changes in hemodynamic response over an extended period, including in a chronic context, or as in the present study that included 4 x 60 min assessments spread over 8 h. 

Light was emitted at 691 and 830 nm by optical fibers attached in pairs to four prisms (8 fibers in total) that were separated from the collector bundle by 2.0, 2.5, 3.0 or 3.5 cm. Each of the emitter and collector bundle prisms were embedded into a flexible polyurethane resin to form a sensor with the overall dimensions of 7.6 × 2.5 × 0.3 cm. Identical sensors were attached to the forehead above each eye of participants with medical tape and secured in place with a self-adhering bandage. The sensors were positioned so that the bottom edge was level with the top of the participants’ eyebrows and the middle edge touching at the midline of the forehead. Data were collected at a rate of 5 Hz. Each pair collected data from an area of prefrontal cortex that included the areas corresponding to the international 10–20 system Fp1 and Fp2 EEG positions. 

Given the exploratory nature of the study, several primary endpoints were identified including cognitive function and fatigue during extended performance of the Cognitive Demand Battery at 1, 3 and 6 h post-consumption and long-term declarative memory at 1, 3 and 6 h post-intervention. All other outcomes were identified as secondary endpoints.

### 2.5. Procedure

Participants were required to attend the laboratory on five occasions—a screening visit and four test visits. The screening visit comprised briefing on requirements of the study and signing of the informed consent form, review of the inclusion and exclusion criteria, self-reported health screening, collection of demographic data and training on the cognitive and mood measures. The first test visit was at least 48 h and less than 28 days after the screening visit, and each test visit was separated by a minimum of seven days. On each of the four test visits, participants attended the laboratory at either 07:15 or 08:10 having abstained from all food and drink except water for 12 h, caffeine for 18 h and alcohol for 24 h. In the 24 h prior to each of the test visits, participants were also required to abstain from all polyphenol-containing foods including non-alcoholic wine and non-alcoholic beer, decaffeinated tea and coffee, all fruits and vegetables or products made with fruit or vegetables, all cocoa-containing products, liquorice, honey, herbs or herbal extracts, soybeans or soy products and nuts. Participants were issued with a 24 h diet diary in which to document their food and beverage intake for the pre-testing day, which also contained standardized instructions on which foods to avoid.

Upon arrival at the laboratory, participants were provided with a standard breakfast (2 × medium slices of white toast and 15 g low-fat spreadable butter containing 36 g carbohydrate, 11 g fat, 7 g protein and 2 g fiber; 1138 kJ/272 kcal) followed by a 30 min rest. Following this, the NIRS headband was fitted and the participant then sat quietly for five min. Participants then completed the first ‘baseline’ cognitive assessment with continuous NIRS recording; NIRS data were not collected during completion of the Bond–Lader mood scales, in-house VAS or POMS. The treatment drink was then consumed within 5 min and then identical—save for the presentation of randomly generated parallel versions of each memory task—cognitive/mood assessments with concomitant NIRS recording were completed at 60, 180 and 360 min post-dose. Participants were provided with a light, standard lunch immediately following the 180 min post-dose assessments (cheese sandwich on white bread, vanilla yogurt and savory cracker snacks containing 60 g carbohydrate, 24 g fat, 26 g protein and 2 g fiber; 2301 kJ/550 kcal,). The low-polyphenol study visit meals were designed to provide minimal nourishment in the absence of the same or similar compounds under investigation. Participants could drink water ad libitum throughout the course of the test visits apart from when participating in the cognitive testing. Upon completion of the study, participants received £170 honorarium or pro rata for withdrawals. Appendix A depicts the schedule of the testing session.

### 2.6. Statistics

#### 2.6.1. Cognitive and Mood Assessment

Linear mixed models were used to test hypotheses using the mixed procedure in IBM SPSS (version 22.0, IBM corp. Portsmouth, UK). Normality of the variables at each assessment was inspected, and variables were log-transformed if skewed; a constant was added to all data for variables which included 0 values. Means ± SD of all outcome variables are presented in Appendix A (Online Appendix A). For the models that included CDB outcomes as the dependent variable (including mental fatigue and alertness), treatment, repetition of task (1 to 4) and time (60, 180, 360 min) were entered into the model as fixed effects along with the interactions between each of these, i.e., treatment × repetition × time; treatment × repetition; treatment × time; repetition × time. Respective pre-dose scores (baseline assessment) were also entered into the model as a covariate and subject was included as a random effect with a variance components covariance structure. The best-fitting covariance structure for the residuals across assessments was an autoregressive structure. Significant main effects of treatment or significant interactions with treatment were analyzed further using pairwise comparisons with Bonferroni corrections for multiple testing with placebo as the reference category. Main and interaction effects were considered statistically significant at *p* < 0.05 and for Bonferroni-adjusted post-hoc tests the *p*-value was set at *p* < 0.05. For all of the remaining cognitive and mood outcomes models were used to fit the post-dose data using the procedure described above. The fixed effects for these models were treatment, time, treatment x time with respective pre-dose scores as a covariate; subject was included as a random effect. 

#### 2.6.2. NIRS Assessment

A single data point was calculated for each four (Serial 3 s, Serial 7 s) or five (RVIP) minute task epoch by averaging all of the data points collected within the epoch. The analysis of the post-dose cerebral blood flow parameters across the entire visit comprised fitting a linear mixed model including treatment, hemisphere (left/right) and epoch (11 × 4/5 min epochs during each assessment = 33 epochs) and their interactions, i.e., treatment × hemisphere × epoch; treatment × hemisphere; treatment × epoch, hemisphere × epoch as fixed effects. Respective pre-dose values (baseline assessment) were also entered into the model as a fixed effect and subject was included as a random effect with a variance components covariance structure. The best-fitting covariance structure for the residuals across assessments was an autoregressive structure. Significant fixed effects were analyzed further using pairwise comparisons as above.

## 3. Results

Of the thirty-five participants who were randomized to receive treatment, three participants withdrew post-randomization—one participant withdrew consent, one was lost to follow up and one was withdrawn by the investigator due to inadequate NIRS data acquisition. All withdrawn participants were replaced in order to achieve a final dataset comprising 32 complete participants. See Appendix A (Online Appendix A) for participant disposition diagram. For the VAS measures (Bond–Lader mood scales, in-house VAS), blind analysis revealed that one participant did not complete the scales according to the instructions by only using the extreme ends of the answer line. These data were subsequently removed, therefore leaving 31 datasets for these outcome measures. Two adverse events were reported as ‘possibly related to the treatment’ (headache, drowsiness; both following the placebo beverage) and one was reported as ‘probably related to the treatment’ (headache following the coffee berry extract beverage), while all the remaining adverse events (*n* = 9) were reported as ‘unrelated to the treatment’. The dietary restrictions were not adhered to by two participants for one study visit each (these two participants ate chocolate or tomatoes in the 24 h prior to their placebo and coffee berry beverage study visits, respectively). As these study visits could not be re-scheduled, these data were entered into the final dataset. Similarly, there were two instances where there were only six days between study visits for the same participant and these data were entered into the final dataset. On this note, the majority of participants completed each visit within seven days of their previous visits (74%); the maximum time between any two study visits was 41 days. 

### 3.1. Baseline Scores

There were no statistically significant differences between treatments at baseline for any of the cognitive or mood measures. However, a main effect of treatment was observed at baseline for StO_2_ [F(3, 314.17) = 11.65, *p* < 0.001] and THb [F(3, 299.45) = 18.89, *p* < 0.001]. For both outcomes, pairwise comparisons revealed higher concentrations in the placebo condition compared to all three active treatments (all *p* < 0.001).

### 3.2. Cognitive and Mood Assessment

#### 3.2.1. Cognitive Performance

Main effects of treatment were detected for total responses for serial 3 subtractions, total responses for serial 7 subtractions, RVIP false alarms and reaction time on the word recognition task (all *p* < 0.05). However, in all instances post-hoc comparisons between the investigational beverages and placebo were not significantly different. A significant main effect of treatment was also observed for number of errors of the delayed word recall task [ F (3, 113.48) = 3.08, *p* = 0.030]. Post-hoc comparisons revealed more errors were made following the coffee berry extract beverage, compared to placebo (*p* = 0.014). However, this was in the absence of any significant difference on the number of words correctly recalled. No other main effects or interactions with treatment were observed.

#### 3.2.2. Mood

A significant main effect of treatment was found for subjective ratings of alertness measured after each repetition of the CDB [F(3, 230.43) = 3.18, *p* = 0.025]. Here, ratings of alertness were significantly higher following the apple extract beverage, compared to placebo (*p* = 0.0499). A significant main effect of treatment was also found for subjective ratings of mental fatigue also measured after each repetition of the CDB [F(3, 225.63) = 5.86, *p* = 0.001]. Ratings of mental fatigue were lower following the apple extract (*p* = 0.002) and coffee berry extract (*p* = 0.003) beverages, compared to placebo. See Figure 2 for VAS graph.

Significant effects of treatment were found for a number of the POMS outcome measures: anger/hostility [F(3, 109.27) = 3.61, *p* = 0.016], confusion/bewilderment [F(3, 132.02) = 5.30, *p* = 0.002], depression/dejection [F(3, 108.77) = 3.46, *p* = 0.019], fatigue/inertia [F(3, 127.09) = 6.46, *p* < 0.001] and total mood disturbance [F(3, 119.06) = 5.75, *p* = 0.001]. Post-hoc comparisons against placebo are shown in Figure 3. These revealed that anger/hostility (*p* = 0.042) and depression/dejection (*p* = 0.013) were reduced following the blueberry extract beverage, and anger/hostility (*p* = 0.009) was reduced following the apple extract beverage. Confusion/bewilderment (*p* = 0.001), fatigue/inertia (*p* < 0.001) and total mood disturbance (*p* < 0.001) were all reduced following the coffee berry extract beverage.

#### 3.2.3. NIRS 

Total hemoglobin: A significant main effect of treatment was found for tHb [F(3, 2749.47) = 4.87, *p* = 0.002], with increased concentration of tHb observed following the apple extract beverage across all three post-dose assessments (*p* = 0.001), compared to placebo. In addition, a significant treatment x hemisphere interaction was observed for tHb [F(3, 2190.16) = 7.82, *p* < 0.001]. Post-hoc comparisons revealed that, compared to placebo, tHb was increased following the apple (*p* = 0.030) and coffee berry (*p* < 0.001) extract beverages in the right hemisphere, and reduced in the left hemisphere following the coffee berry extract beverage (*p* = 0.046). 

Oxygen saturation: A significant main effect of treatment for StO_2_ was found [F(3, 2256.44) = 25.78, *p* < 0.001], with increased oxygen saturation being observed following the blueberry (*p* < 0.001), apple (*p* < 0.001) and coffee berry (*p* = 0.001) extract beverages across all three post-dose assessments, compared to placebo. A significant treatment x hemisphere interaction was also observed here. Compared to placebo, StO_2_ was increased in the right hemisphere following the blueberry and apple extract beverages (both *p* < 0.001) and increased in the left hemisphere following all of the active beverages (all *p* < 0.001). 

The above NIRS outcomes are represented graphically in Figure 4. There were no interactions between treatment and either assessment or epoch on any NIRS measure.

## 4. Discussion

The current study aimed to investigate the effect of combinations of differing phenolic-rich extracts with other phytochemicals, in the form of beetroot, ginseng and sage, on CBF, cognitive function and mood. The results here demonstrate that the combination of beetroot, sage, ginseng and phenolics from different structural groups, in the form of flavanol-rich apple extract, anthocyanin-rich blueberry extract or phenolic acid-rich coffee berry extract, did indeed lead to differential patterns of modulation of mood and CBF parameters in healthy humans. The most striking finding was a clear and consistent improvement in psychological state across the different mood measures following both the phenolic acid- and flavanol-rich beverages. All three active treatments also led to increased oxygen saturation in the frontal cortex, but only the flavanol-rich extract lead to increased CBF per se (as indexed by total levels of hemoglobin).

If we look first at the mood effects, the phenolic acid-rich coffee berry extract beverage, in comparison to placebo, was associated with reduced mental fatigue ratings during completion of the CDB and reduced fatigue/inertia post-assessment. In addition, participants reported a decrease in confusion/bewilderment and total mood disturbance. These findings might, on the surface, be more readily attributed to the 22 mg caffeine also contained within this extract but several factors might counterweight this argument. Firstly, while some will argue that this attenuation of fatigue is the result of reversing the putative effects of caffeine withdrawal caused by the imposed 18 h caffeine abstinence prior to each testing session [42], the evidence for any contribution of withdrawal to caffeine’s psychoactive effects is weak [43,44], particularly following the short period of abstinence here [45]. Secondly, it should be noted in any event that the dose of caffeine here (22 mg) was below the threshold dose (35 mg) generally believed to engender any psychoactive effects, even in overnight abstinent participants [43]. Thirdly, a similar trial conducted at approximately the same time as this one found that a decaffeinated coffee berry extract was also capable of attenuating perceptions of increased fatigue and reduced alertness following similarly cognitively demanding tasks. However, it is worth noting that while the cohort was equivalent to that reported here (*N* = 30, 19–47 yrs males and females), the doses are not—being 300 and 100 mg in this latter trial.

Nevertheless, there is sufficient evidence for the alerting effects of decaffeinated chlorogenic acid (CGA)-rich extract from other sources [46,47] to support its ability here. The latter of these studies [46] showed an acute alerting effect (as well as improved cognitive performance) following a high CGA decaffeinated coffee containing 5 mg caffeine that was not seen following a similar dose of pure CGA/caffeine, suggesting that other constituents in the coffee-based drink contributed to these effects see also [48]. Further, importantly, the effects in the current study were seen as a main effect across all three post-dose assessments (60, 180, 360 min). Given the extended time course of effects seen here, the similarity of findings on alertness across other studies, and the lack of such effects from other studies assessing caffeine alone at this dose, it is unlikely that caffeine is the main active constituent underlying the results seen here. However, it should also be noted that co-consumption of comparatively low doses of caffeine has also been shown to increase the bioavailability of phenolic compounds [49,50] and (along with other methylxanthines) exert a synergistic influence on the cardiovascular effects of cocoa-flavanols [50]. As such, a similar interactive contribution to the psychological effects of coffee berry extract seen here cannot be ruled out. 

The apple extract beverage was also associated with a number of beneficial effects on mood. Two patterns emerged from the data. Firstly, during completion of the CDB, subjective ratings of alertness were increased and mental fatigue decreased following the apple extract beverage, compared to the control drink. This finding is mirrored with cocoa flavanols, a class of polyphenols within the same flavonoid family as the catechins found in the apple extract [4]. Secondly, the apple extract beverage was associated with reduced anger/hostility; indicative of an anxiolytic effect of the apple extract beverage. In mice, (-)epicatechin administration had a similar anxiolytic effect which was associated with modulation of cortical monoamine oxidase-A (MAO-A) levels and hippocampal brain-derived neurotrophic factor [51]. Whether the same mechanism underlies the current, acute effect in humans remains to be investigated.

The beneficial psychological effects associated with the anthocyanin-rich drink included a reduction in anger/hostility and depression/dejection on the POMS. Given that all three phenolic-rich extracts were added to an identical combination of beetroot, sage and ginseng, the differential effects on mood suggest one of two possibilities. The first is that the effects are driven by the individual phenolic extracts themselves, and previous research that has administered similar compounds would support this hypothesis [4,46,47,52,53]. The alternative is that the specific phenolic compounds contained within each extract interacted with any one of the other extracts differently and possibly synergistically. For example, although not fully understood, the mechanism of action of sage is thought to be due to its ability to inhibit cholinesterase activity [24], a mechanism also posited for numerous polyphenolic compounds. However, the mechanisms by which this occurs are dependent on the structural specificity of the phenolic compound in question [54]. In addition, different phenolic compounds have been shown to have differential efficacy with regards to modulation of NO metabolism following the consumption of nitrate [18]. 

A limitation of the current study must be noted here— a beverage containing only the beetroot, sage and ginseng extracts was not included as a comparator and, therefore, conclusions regarding the underlying mechanisms of the effects seen here are difficult to draw. In future, for clarity, this comparator drink would be included to help partial out the potential input that actives like betalains (beetroot) and ginsensosides (ginseng) may have had on the study outcomes. While no mood effects have, to the best of current knowledge, been attributed to betalins in humans (aside from some very recent effects related to perceived exertion in athletes [55]), they are certainly capable of modulating cerebral blood flow in healthy adults [21]. A recent systematic review also highlighted the benefits of ginseng on the circulatory/cardiovascular system (in part related to modulating blood flow), but, again, no reported effects on mood [56], suggesting that beetroot and ginseng were unlikely to have impacted mood here but that they were certainly capable of modulating blood flow.

In terms of CBF effects, all of the investigational beverages elicited an increase in oxygen saturation in the frontal cortex during tasks that activate this brain region [14,21,38]. However, only the flavanol-rich apple extract beverage resulted in increased levels of total hemoglobin—a proxy measure of increased CBV—and may underpin the above findings on subjective alertness/fatigue, as has been suggested in the case of cocoa flavanols [4]. The former, universal effect on oxygen saturation suggests that a component or group of components common to all three beverages may have contributed to this effect. As discussed above, the methodology here did not allow the exclusion of the three phenolic extracts as potentially underlying this effect and the beetroot, sage and ginseng extracts are all likely candidates given their known modulation of blood flow and NO synthesis [12,26,27,28]. 

The hemispheric interaction effect observed for total hemoglobin is more difficult to explain. The most obvious suggestion is that, given the cognitive tasks utilized here are sub-served most strongly by the left hemisphere, e.g., [57,58], the flavanol-rich apple extract intervention is merely amplifying a natural, NO-driven increase in CBV in response to increased demand. Why the coffee berry extract should attenuate this is most likely attributed to the small amount of caffeine (22 mg) present in this extract. As discussed above, there is little evidence that caffeine is capable of evincing cognitive/mood effects at this relatively low dose but nothing to indicate that it cannot still exert physiological effects, including reducing CBV. It must be noted that the NIRS technique itself is limited by its poor spatial resolution in comparison to other neuroimaging techniques [e.g., EEG and fMRI [59]] and so hemispheric/regional differences in response to treatment using NIRS could be investigated further with the addition of more channels. In addition, the presence of baseline differences indicating initial higher concentrations of StO_2_ and THb in the placebo condition compared to all the three active treatments suggests that the above findings should be interpreted with caution. Indeed, the observed post-dose effects may have been attenuated in spite of the inclusion of baseline scores in the analysis.

It is perhaps surprising that no interpretable cognitive effects were noted given the observed modulation of CBF parameters discussed above, especially with regard to the apple extract beverage. The same pattern of increased CBV, in the absence of cognitive effects has been seen with other polyphenols using the same methodology [14,31,53] and might suggest that this effect is predicated on methodological factors. For example, the dose of catechins here (~235 mg) may have been too low, the restrictive NIRS headgear too oppressive [37] or the CBF parameter effects too small to elicit effects in young populations. Here, older populations, where oxygen delivery and mitochondrial function are in decline [60], may present a more sensitive cohort to investigate. 

In summary, it was hypothesized that the addition of three different structural groups of phenolics (anthocyanins, flavanols and phenolic acids) to a botanical drink containing potentially bioactive [61,62,63] beetroot, sage and ginseng would lead to differential patterns of modulation of cognitive function, mood and CBF parameters. All three active drinks demonstrated varying degrees of modulation of mood and CBF which suggests perhaps that the beetroot, sage and ginseng combination was amplified by the addition of phytochemical polyphenols. It is of course equally possible that the polyphenol extracts were solely responsible for the effects seen and this may explain the lack of overlap in the observed effects between each beverage. In conclusion, the combination of different phytochemical extracts may have potentially beneficial effects on brain function, but the specific synergies between active constituent compounds remain to be explored.

## Figures and Tables

**Figure 1 nutrients-12-02254-f001:**
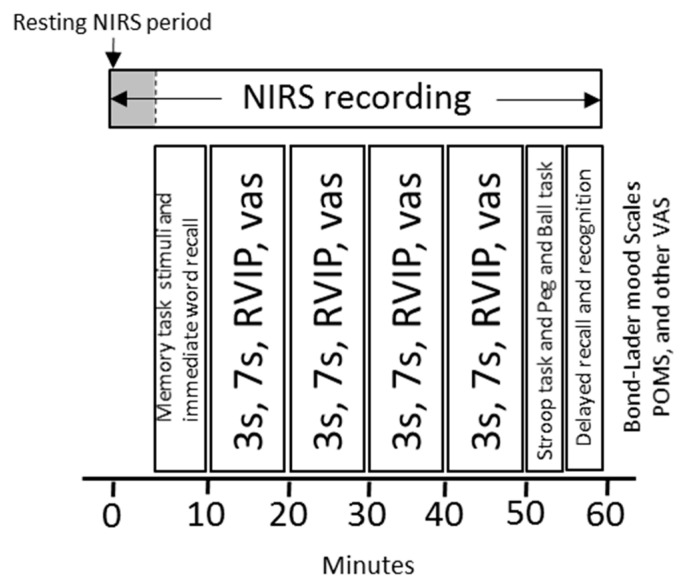
Cognitive and mood assessment with concurrent NIRS recording. 3 s, serial 3 subtractions; 7 s, serial 7 subtractions; RVIP, rapid visual information processing; VAS, visual analogue scales; POMS, Profile of Mood States questionnaire.

**Figure 2 nutrients-12-02254-f002:**
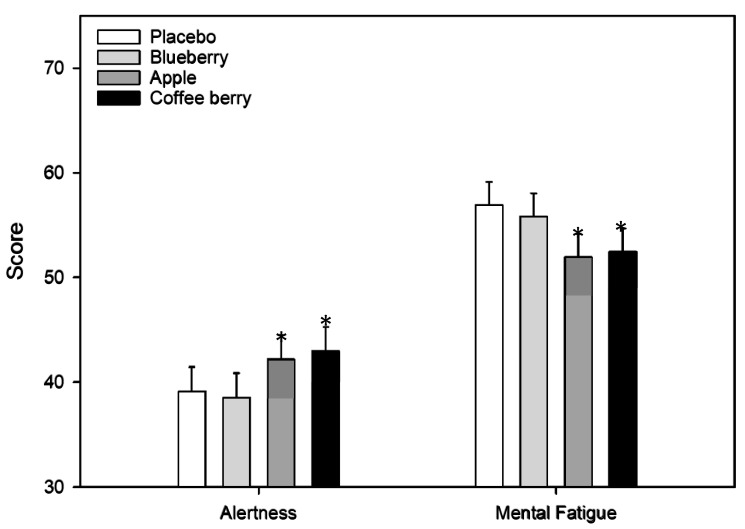
Visual analogue scales. Alertness and mental fatigue were measured during completion of the Cognitive Demand Battery. Values are adjusted means ± SEM derived from the linear mixed model using untransformed data (*n* = 32). Treatment, repetition of task (1 to 4) and time (60, 180, 360 min) were entered into the model as fixed effects along with the interactions between each of these, i.e., treatment × repetition × time; treatment × repetition; treatment × time; repetition × time. Respective pre-dose scores (baseline assessment) were also entered into the model as a covariate and subject was included as a random effect. Effects were calculated using log-transformed data. Post-hoc comparisons were made between placebo and each of the active treatments using a Bonferroni adjustment for multiple comparisons. *, *p* < 0.05 (post-hoc test against placebo).

**Figure 3 nutrients-12-02254-f003:**
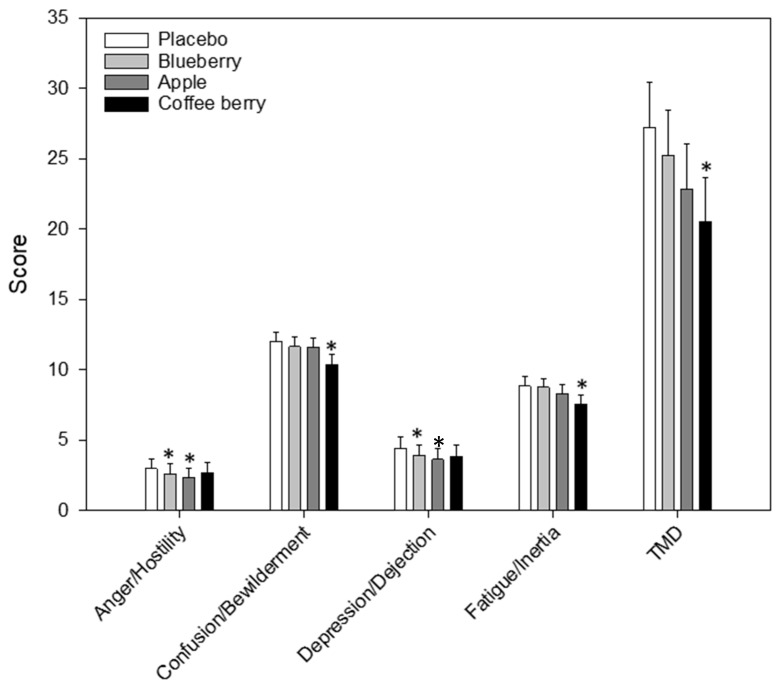
Profile of mood states. Values are adjusted means ± SEM derived from the linear mixed model using untransformed data (*n* = 32). Treatment and time (60, 180, 360 min) and their interaction, i.e., treatment x time were entered into the model as fixed effects. Respective pre-dose scores (baseline assessment) were also entered into the model as a covariate and subject was included as a random effect. Effects were calculated using log-transformed data. Post-hoc comparisons were made between placebo and each of the active treatments using a Bonferroni adjustment for multiple comparisons. *, *p* < 0.05 (post-hoc test against placebo).

**Figure 4 nutrients-12-02254-f004:**
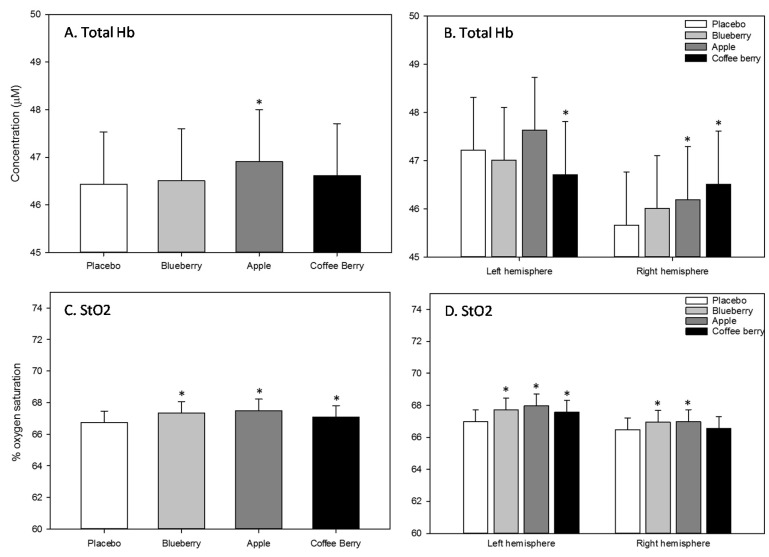
Near-infrared spectroscopy (NIRS) outcomes. Significant main effects of treatment (**A**. Total Hb and **C**. StO2) and significant treatment x hemisphere interactions (**B**. Total Hb and **D**. StO2) were observed for all outcomes (*n* = 32). Values are adjusted means ± SEM derived from the linear mixed model. Treatment, hemisphere (left/right) and epoch (11 × 4/5 min epochs during each assessment = 33 epochs) and their interactions, i.e., treatment × hemisphere × epoch; treatment × hemisphere; treatment × epoch, hemisphere × epoch were entered into the models as fixed effects. Respective pre-dose values (baseline assessment) were also entered into the model as a covariate and subject was included as a random effect. Post-hoc comparisons were made between placebo and each of the active treatments using a Bonferroni adjustment for multiple comparisons. *, *p* < 0.05 (post-hoc test against placebo); Total-Hb, total hemoglobin; StO_2_, oxygen saturation.

**Table 1 nutrients-12-02254-t001:** Description of the investigational beverages. All beverages were made up to 10 fl oz.

Placebo	Blueberry	Apple	Coffee Berry
Base ^1^	Base ^1^	Base ^1^	Base ^1^
	Blueberry extract ^2^	Apple extract ^6^	Coffee berry extract ^7^
	Beetroot extract ^3^	Beetroot extract ^3^	Beetroot extract ^3^
	Ginseng extract ^4^	Ginseng extract ^4^	Ginseng extract ^4^
	Sage extract ^5^	Sage extract ^5^	Sage extract ^5^

^1^ Water (95%), sucralose (4%), preservatives, and artificial flavors (1%). ^2^ Bluberry extract (2.49 g)—North American blueberry extract standardized to contain a minimum of 12% blueberry anthocyanins. Supplied by Futureceuticals, United States of America (USA). ^3^ Beetroot extract (10 g)—a fresh beetroot juice powder standardized for 1.5% nitrate and 0.4% betalains. Supplied by Futureceuticals, USA. ^4^ Ginseng (170 mg)—ginseng extract standardized to contain 4.5% ginsenosides. ^5^ Sage extract (280 mg)—powdered sage extract. ^6^ Apple extract (275 mg)—standardized to contain 85% polyphenols expressed as catechin equivalents [(-)-epicatechin min. 30% w/w; flavan-3-ol oligomers min. 20% w/w]. Supplied by Correscence Ltd., United Kingdom (UK). ^7^ Coffee berry extract (1.1 g)—an extract of powdered whole coffee fruit standardized to contain 40% chlorogenic acids and 2% caffeine. Supplied by Futureceuticals, USA.

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
