# Peer review of "A Randomized, Crossover Study of the Acute Cognitive and Cerebral Blood Flow Effects of Phenolic, Nitrate and Botanical Beverages in Young, Healthy Humans"

_nutrients, 2020, doi:10.3390/nu12082254_

Round 1

Reviewer 1 Report

The manuscript entitled "A randomized, crossover study of the acute cognitive and cerebral blood flow effects of phenolic, nitrate and botanical beverages in young, healthy humans" is an interesting study that will contribute to the current literature. However, it needs to be revised. Some recommendations are as the following:

- The major problem is that it is not clear how the polyphenol extractions are carried out or how the polyphenol contents of the beverages are determined. Please give the details of the methods that are used to extract and measure the polyphenols from blueberry, apple and coffee berry.

- It was hypothesized that the observed beneficial effects are related to polyphenols. However, there are several other compounds present in the beverages that might contribute to the observed effects. The authors discussed this issue only in relation to caffeine. How about compounds such as betalains or ginsenosides?

- Another major issue is the lack of control (the beverage that only contains beetroot, ginseng and sage extract). It is difficult to draw conclusions without the control.

- In Fig. 2, there should be a star above the bar showing the alertness value of coffee berry, no? Similar situation is also valid for the depression/dejection value of apple extract in Fig 3.

- Lines 395-400: These statements require citation.

Author Response

The manuscript entitled "A randomized, crossover study of the acute cognitive and cerebral blood flow effects of phenolic, nitrate and botanical beverages in young, healthy humans" is an interesting study that will contribute to the current literature. However, it needs to be revised. Some recommendations are as the following:

1. The major problem is that it is not clear how the polyphenol extractions are carried out or how the polyphenol contents of the beverages are determined. Please give the details of the methods that are used to extract and measure the polyphenols from blueberry, apple and coffee berry.

We agree that this information will be beneficial and so have included this in the supplementary information for coffee berry and apple. Unfortunately, due to the time that has elapsed since conducting this trial, we no longer have contact with the company who provided the blueberry extract. We have tried to locate this person but are still waiting on a potential response. We didn’t want to delay the resubmission by holding out for this but it may or may not come to light in time for the publication.

2. It was hypothesized that the observed beneficial effects are related to polyphenols. However, there are several other compounds present in the beverages that might contribute to the observed effects. The authors discussed this issue only in relation to caffeine. How about compounds such as betalains or ginsenosides?

This has now been addressed in lines 403-411 of the discussion.

3. Another major issue is the lack of control (the beverage that only contains beetroot, ginseng and sage extract). It is difficult to draw conclusions without the control.

This is definitely an issue and we’ve partialed this out of the paragraph it originally ended (was maybe a bit of a throwaway comment here) so that it now begins its own paragraph (lines 400-402) and this now logically leads on to discussion of the potential role of betalains and ginsenosides (lines 403-411) which you highlight above.

4. In Fig. 2, there should be a star above the bar showing the alertness value of coffee berry, no? Similar situation is also valid for the depression/dejection value of apple extract in Fig 3.

Stars are now added where highlighted for both figures. Thank you.

5. Lines 395-400: These statements require citation.

These citations for left-hemisphere task lateralization have now been added to line 424.

Reviewer 2 Report

Hello Authors

Very thorough study on the phytochemical combinations and its effects on mood and other cognitive parameters.

I will suggest to add thin layer chromatograms of all the botanical extracts used; to see if there are similar phytochemicals among them attributing to synergistic effects.

Author Response

Very thorough study on the phytochemical combinations and its effects on mood and other cognitive parameters.

6. I will suggest to add thin layer chromatograms of all the botanical extracts used; to see if there are similar phytochemicals among them attributing to synergistic effects.

The information relating to extraction and composition is now included in the supplemental information for coffee berry and apply. Unfortunately, as above, despite our best efforts we weren’t able to source this information for the blueberry extract.

Reviewer 3 Report

Although the study seems to be interesting and has scientific potential, the entire conclusion is observation based and no molecular or cellular or biochemical experiments were performed to investigate the underlying mechanisms. This makes the entire study observational and less meaningful. Additionally, the authors need to report the phytochemical compositions of the blueberry, apple and coffee berry that were used in the study.

Author Response

7. Although the study seems to be interesting and has scientific potential, the entire conclusion is observation based and no molecular or cellular or biochemical experiments were performed to investigate the underlying mechanisms. This makes the entire study observational and less meaningful. Additionally, the authors need to report the phytochemical compositions of the blueberry, apple and coffee berry that were used in the study.

The information relating to composition is now included in the supplemental information but, unfortunately, this is not available for the blueberry extract.

Round 2

Reviewer 1 Report

The authors revised the manuscript considering the majority of the reviewer’s comments. However, there are still a few minor points that needs to be addressed before the paper can be accepted for publication in "Nutrients".

- In the response letter, the authors indicated that they could not include the details of polyphenol content and extraction method of blueberry. Although it is not possible to revise the manuscript after publication, perhaps it might be possible to update the supplementary material. Therefore, I strongly suggest authors to include this information to the supplementary material file once they receive it.

- Again, the authors indicated that they revised the statistical analysis results (i.e., stars above bars) in Fig 2 and Fig 3. However, they are the same figures presented in the previous version. Please check this once more.

Author Response

  1. Yes, that's a good idea. I'll request with Nutrients that the blueberry content be updated once it becomes available in order not to delay publication within this special issue.
  2. Apologies, I hadn't realised that the stars had not transferred when copying and pasting the amended figures. That should be correct now.

Reviewer 2 Report

Changes have been made.

Author Response

Thank you.